# CD10-Bound Human Mesenchymal Stem/Stromal Cell-Derived Small Extracellular Vesicles Possess Immunomodulatory Cargo and Maintain Cartilage Homeostasis under Inflammatory Conditions

**DOI:** 10.3390/cells12141824

**Published:** 2023-07-11

**Authors:** Dimitrios Kouroupis, Lee D. Kaplan, Johnny Huard, Thomas M. Best

**Affiliations:** 1Department of Orthopaedics, UHealth Sports Medicine Institute, University of Miami Miller School of Medicine, Miami, FL 33146, USAtxb440@med.miami.edu (T.M.B.); 2Diabetes Research Institute & Cell Transplant Center, University of Miami Miller School of Medicine, Miami, FL 33136, USA; 3Linda and Mitch Hart Center for Regenerative and Personalized Medicine, Steadman Philippon Research Institute, Vail, CO 81657, USA; jhuard@sprivail.org

**Keywords:** mesenchymal stem/stromal cells (MSC), synovium, infrapatellar fat pad (IFP), CD10 (neprilysin), PRG4 (lubricin), small extracellular vesicles (sEVs), immunomodulation, pain, chondroprotection, inflammatory joint diseases

## Abstract

The onset and progression of human inflammatory joint diseases are strongly associated with the activation of resident synovium/infrapatellar fat pad (IFP) pro-inflammatory and pain-transmitting signaling. We recently reported that intra-articularly injected IFP-derived mesenchymal stem/stromal cells (IFP-MSC) acquire a potent immunomodulatory phenotype and actively degrade substance P (SP) via neutral endopeptidase CD10 (neprilysin). Our hypothesis is that IFP-MSC robust immunomodulatory therapeutic effects are largely exerted via their CD10-bound small extracellular vesicles (IFP-MSC sEVs) by attenuating synoviocyte pro-inflammatory activation and articular cartilage degradation. Herein, IFP-MSC sEVs were isolated from CD10High- and CD10Low-expressing IFP-MSC cultures and their sEV miRNA cargo was assessed using multiplex methods. Functionally, we interrogated the effect of CD10High and CD10Low sEVs on stimulated by inflammatory/fibrotic cues synoviocyte monocultures and cocultures with IFP-MSC-derived chondropellets. Finally, CD10High sEVs were tested in vivo for their therapeutic capacity in an animal model of acute synovitis/fat pad fibrosis. Our results showed that CD10High and CD10Low sEVs possess distinct miRNA profiles. Reactome analysis of miRNAs highly present in sEVs showed their involvement in the regulation of six gene groups, particularly those involving the immune system. Stimulated synoviocytes exposed to IFP-MSC sEVs demonstrated significantly reduced proliferation and altered inflammation-related molecular profiles compared to control stimulated synoviocytes. Importantly, CD10High sEV treatment of stimulated chondropellets/synoviocyte cocultures indicated significant chondroprotective effects. Therapeutically, CD10High sEV treatment resulted in robust chondroprotective effects by retaining articular cartilage structure/composition and PRG4 (lubricin)-expressing cartilage cells in the animal model of acute synovitis/IFP fibrosis. Our study suggests that CD10High sEVs possess immunomodulatory miRNA attributes with strong chondroprotective/anabolic effects for articular cartilage in vivo. The results could serve as a foundation for sEV-based therapeutics for the resolution of detrimental aspects of immune-mediated inflammatory joint changes associated with conditions such as osteoarthritis (OA).

## 1. Introduction

In inflammatory joint diseases, including certain phenotypes of osteoarthritis (OA), the synovium and IFP tissues serve as origins of pro-inflammatory and articular cartilage degradative molecules, as well as being a source of the pain-transmitting immune and inflammation modulator neuropeptide substance P (SP) [1,2,3,4,5,6,7,8]. In fact, OA-related knee pain may be driving the association with time to mortality [9]. Given the current challenges of identifying disease-modifying treatment strategies for patients with OA [10], novel alternatives such as MSC-based therapeutic approaches have yielded encouraging initial clinical results. Early-stage clinical trials using freshly isolated or culture-expanded MSC derived mainly from bone marrow and adipose tissues have demonstrated clinical superiority when compared with current alternatives such as hyaluronic acid intra-articular placement [11]. Therefore, to reverse these detrimental pathological cascades, we originally focused on MSC-based immunomodulatory therapies, as their paracrine effects actively modulate immune, inflammatory, and fibrotic events [12,13,14,15]. We recently reported that human IFP-MSC, after transiently engrafting into joint areas of active synovitis/IFP fibrosis, show a potent anti-inflammatory/analgesic phenotype by actively degrading SP via neutral endopeptidase CD10 [16,17].

Concern regarding the potential side effects and increased immunogenicity with MSC-based therapies has led researchers to pursue cell-free treatments focusing on MSC secretome, especially their small extracellular vesicles (sEVs). sEVs are nanosized (50–200 nm) extracellular vesicles generated via the endosomal pathway [18], and secreted by numerous cells in response to their surrounding milieu. Thus, their contents (i.e., cargo) and lipid shell may carry information that reflects particular changes in the parent cells. According to our data, the in vitro SP degradation capacity of IFP-MSC can be fully recapitulated by their supernatant alone, suggesting the release of active sEV-bound CD10 from the cells [17]. On this basis, further investigations revealed that IFP-MSC-derived sEVs (IFP-MSC sEVs) show distinct miRNA and protein immunomodulatory profiles. Specifically, IFP-MSC sEVs’ infusion into the knee in an acute synovial/IFP inflammation rat model resulted in robust macrophage polarization towards an anti-inflammatory therapeutic M2 phenotype within the synovium/IFP tissues [19]. Overall, preclinical studies have demonstrated that MSC sEVs have strong immunomodulatory properties, particularly through the action of miRNAs, which may be capable of targeting the immune system and modulating angiogenesis [10].

We and others have isolated and characterized sEVs from various MSC sources (i.e., bone marrow, umbilical cord, adipose tissues, endometrium). Our results support a therapeutic potential through robust anti-inflammatory, anti-fibrotic, and angiogenesis-remodeling capacities [19,20,21]. Our hypothesis is that CD10 expression levels in IFP-MSC are directly related to the immunomodulatory and chondroprotective effects of their derived sEVs. On this basis, CD10High sEVs show a potent molecular immunomodulatory profile which significantly affects synoviocytes’ functionality in inflammatory conditions. Importantly, chondroprotective effects of CD10High sEVs were not only observed in chondropellet/synoviocyte cocultures under inflammatory conditions in vitro, but similarly in an animal model of acute synovitis/IFP fibrosis by retaining articular cartilage structure/composition. These observations provide a rationale for further testing of a viable MSC sEV-based therapeutic modality for synovitis/IFP fibrosis as well as chronic conditions such as OA, where articular cartilage degradation is a critical component of the disease.

## 2. Materials and Methods

### 2.1. Isolation, Culture, and Expansion of IFP-MSC

All experiments using human cells were performed in accordance with relevant guidelines and regulations. Human IFP-MSC were isolated from IFP tissue obtained from de-identified, non-arthritic patients (n = 3; two males 46 and 67 years old, and one female 44 years old) undergoing elective knee arthroscopy at the Lennar Foundation Medical Center at the University of Miami. Informed consent was obtained from all participants. All procedures were carried out following approval by the University of Miami IRB as non-human research (based on the nature of the samples as discarded tissue). IFP tissue (5–10 cc) was mechanically dissected and washed repeatedly with Dulbecco’s phosphate buffered saline (DPBS; Sigma Aldrich, St. Louis, MO, USA), followed by enzymatic digestion using 235 U/mL collagenase I (Worthington Industries, Columbus, OH, USA) diluted in DPBS and 1% bovine serum albumin (Sigma) for 2 h at 37 °C with agitation. Enzymatic digestion was inactivated with complete media with DMEM low glucose (1 g/L) GlutaMAX (ThermoFisher Scientific, Waltham, MA, USA) containing 10% fetal bovine serum (FBS; VWR, Radnor, PA, USA), washed and seeded at a density of 1 × 10^6^ cells/175 cm^2^ flask in two different complete media: human platelet lysate (hPL) and chemically reinforced (Ch-R) media. Complete hPL medium was prepared by supplementing DMEM low-glucose GlutaMAX with 10% hPL solution (PL Bioscience, Aachen, Germany). Complete Ch-R medium was prepared by mixing mesenchymal stem cell growth medium 2 with supplement provided according to the manufacturer’s instructions (PromoCell, Heidelberg, Germany). At 48 h post-seeding, non-adherent cells were removed by DPBS rinsing and fresh media were replenished accordingly.

All MSC were cultured at 37 °C, 5% (*v*/*v*) CO_2_ until 80% confluent as passage 0 (P0), then passaged at a 1:5 ratio until P3, detaching them with TrypLE™ Select enzyme 1X (Gibco, ThermoFisher Scientific) and assessing cell viability with 0.4% (*w*/*v*) trypan blue (Invitrogen, ThermoFisher Scientific). Specifically, Ch-R and hPL IFP-MSC cultures yielded CD10High and CD10Low IFP-MSC, respectively.

### 2.2. Immunophenotype

Flow cytometric analysis was performed on P3 IFP-MSC (n = 3). Briefly, 2.0 × 10^5^ cells were labelled with CD10 monoclonal antibody (Biolegend, San Diego, CA, USA) and the corresponding isotype control. All samples included a ghost red viability dye (Tonbo Biosciences, San Diego, CA, USA). Data (20.000 events collected) were acquired using a Cytoflex S (Beckman Coulter, Brea, CA, USA) and analyzed using Kaluza analysis software 2.1. (Beckman Coulter).

### 2.3. Quantitative Real-Time PCR (qPCR)

RNA extraction was performed using the RNeasy Mini Kit (Qiagen, Frederick, MD, USA) according to the manufacturer’s instructions. Total RNA (1 μg) was used for reverse transcription with a SuperScript™ VILO™ cDNA synthesis kit (Invitrogen), and 10 ng of the resulting cDNA was analyzed by qPCR using a QuantiFast SYBR Green qPCR kit (Qiagen) and a StepOne real-time thermocycler (Applied Biosystems, Foster City, CA, USA). For each target, human transcript primers were selected using PrimerQuest (Integrated DNA Technologies, San Jose, CA, USA) (Appendix A). All samples were analyzed in triplicate. Mean values were normalized to GAPDH, and expression levels were calculated using the 2^−ΔΔCt^ method and represented as the relative fold change of the primed cohort to the naïve (=1).

A predesigned 90 gene Taqman-based MSC qPCR array (Stem Cell Technologies, Appendix A) was performed (n = 2) using 1000 ng cDNA per IFP-MSC sample and processed using a StepOne real-time thermocycler (Applied Biosystems). Data analysis was performed using a qPCR online analysis tool (Stem Cell Technologies). Sample and control Ct values were expressed as 2^−ΔΔCt^ (with 38-cycle cutoff point). The expression levels were represented in bar plots ranked by transcript expression levels on a log-transformed scale of the sample compared to control cohorts. Bar plots were color-coded by the functional class of genes (namely Stemness, MSC, MSC-related/angiogenic, chondrogenic/osteogenic, chondrogenic, osteogenic, adipogenic). CD10High and CD10Low IFP-MSC groups were compared and presented in bar plots.

### 2.4. Isolation and Validation of IFP-MSC-Derived sEVs

CD10High and CD10Low IFP-MSC-derived sEVs were isolated from IFP-MSC-conditioned media by a stepwise ultracentrifugation method and CD63-immunomagnetic purification. Briefly, conditioned media from IFP-MSC groups cultured in sEV-depleted Ch-R or sEV-depleted hPL media [22] were filtered through a 0.22 µm filter to remove debris and large vesicles, and differentially centrifuged for 2000× *g* for 10 min, 10,000× *g* for 30 min, and ultracentrifuged for 120,000× *g* for 16 h [23]. Pre-enriched sEV preparations were incubated with the Dynabeads^®^-based Exosome-Human CD63 Isolation/Detection Reagent (Invitrogen), and using a magnetic separator, sEV preparations were further purified. Samples from each group were assessed for biophysical and biochemical characterization [19].

A CD10 SimpleStep ELISA kit (Abcam, MA, USA) was used to quantify the CD10 protein cargo levels (pg/mL) in IFP-MSC sEVs, following the manufacturer’s instructions. Levels were determined by measuring the fluorescence (450 nm) of individual samples in end-point mode (SpectraMax M5 spectrophotometer, Molecular Devices, San Jose, CA, USA). CD10 levels were normalized to total protein content per group.

The functional assessment of IFP-MSC sEVs was performed in a concentration corresponding to sEVs secreted from 1 × 10^6^ IFP-MSC. For IFP-MSC sEV tracking, sEVs were stained with a PKH26 red fluorescent membrane staining kit (Fluorescent Cell Linker Kits, Sigma) according to the manufacturer’s instructions and co-cultured with target cells in functional assessments.

### 2.5. miRNA Profile of IFP-MSC sEVs

miRNA was extracted from CD10High and CD10Low sEVs using a Total Exosome RNA and Protein Isolation Kit (Thermo Fisher Scientific) according to the manufacturer’s instructions. Total sEV miRNA (1 μg) was used for first-strand cDNA synthesis with an All-in-One miRNA First-Strand cDNA Synthesis Kit (GeneCopoeia, Rockville, MD, USA).

Predesigned human MSC exosome 166 miRNA qPCR arrays (GeneCopoeia) were performed using 1000 ng cDNA per IFP-MSC sample (n = 2), and processed using a StepOne real-time thermocycler (Applied Biosystems, LLC). Data analysis was performed using qPCR results with GeneCopoeia’s online data analysis system (http://www.genecopoeia.com/product/qpcr/analyse/, accessed on 10 March 2023). Mean values were normalized to small nucleolar RNA, C/D box 48 (SNORD48), and expression levels were calculated using the 2^−ΔCt^ method. Putative miRNA interactomes were generated using an miRNet-centric network visual analytics platform (https://www.mirnet.ca/, accessed on 20 March 2023). The miRNA target gene data were collected from the well-annotated database miRTarBase v8.0 and miRNA-gene interactome network refining was performed with a 2.0 betweenness cutoff. Values (with a 34-cycle cutoff point) were represented in a topology miRNA-gene interactome network using a force atlas layout and hypergeometric test algorithm. The miRDB online database (http://mirdb.org, accessed on 2 April 2023) for prediction of functional miRNA targets was used to correlate highly expressed target genes in macrophages and synoviocytes with specific miRNAs identified by IFP-MSC sEV miRNA profiling. MirTarget prediction scores were in the range of 0–100% probability, and candidate transcripts with scores ≥ 50% are presented as predicted miRNA targets in miRDB [24].

### 2.6. Synoviocyte Inflammation Assay

Passage 1 synoviocytes (SYNs) were expanded in synoviocyte medium (ScienCell, Carlsbad, CA, USA). Synoviocyte/sEV (SYN/IFP-MSC sEVs) cocultures were performed using CD10High and CD10Low sEVs for each sample (n = 2). Cocultures were fed with synoviocyte medium + TIC inflammatory/fibrotic cocktail (15 ng/mL TNFα, 10 ng/mL IFNγ, 10 ng/mL CTGF) for 72 h.

CCK-8 cytotoxicity assay (Cell Counting Kit-8, Sigma) was performed in SYN TIC and IFP-MSC sEV/SYN TIC according to the manufacturer’s instructions. CD10High and CD10Low sEV cytotoxicity was determined by measuring optical densities of individual wells at 450 nm (SpectraMax M5 spectrophotometer, Molecular Devices, San Jose, CA, USA).

RNA extraction from SYN TIC cultures was performed using the RNeasy Mini Kit (Qiagen, Frederick, MD, USA) according to the manufacturer’s instructions. Total RNA (1 μg) was used for reverse transcription with a SuperScript™ VILO™ cDNA synthesis kit (Invitrogen). A predesigned 88-gene human synoviocyte array (GeneQuery™ Human Synoviocyte Cell Biology qPCR Array Kit, ScienCell) was obtained using 1000 ng cDNA per culture and processed using a StepOne real-time thermocycler (Applied Biosystems, LLC). Mean values were normalized to the *ACTB* housekeeping gene; expression levels were calculated using the 2^−ΔCt^ method with a 34-cycle cutoff. Values were represented in dot plots as the relative fold change of the CD10High sEVs/SYN TIC or CD10Low sEVs/SYN TIC to SYN TIC (reference sample, 2^−ΔCt^ = X sample/X reference sample).

The functional enrichment analysis was performed using g:Profiler (version e108_eg55_p17_9f356ae) with the g:SCS multiple testing correction method, applying a significance threshold of 0.05 [25]. The colors for different evidence codes and for log scale are described in Appendix A.

### 2.7. Chondropellets/Synoviocytes Co-Culture Assay

Chondrogenic differentiation (0.25 × 10^6^ IFP-MSC/pellet) was induced for 15 days with serum-free MesenCult-ACF differentiation medium (STEMCELL Technologies Inc., Vancouver, BC, Canada). Chondropellets were harvested and chondropellet/synoviocyte transwell cocultures were obtained with and without CD10High sEVs (n = 2). Cocultures were fed with synoviocyte medium + TIC inflammatory/fibrotic cocktail (15 ng/mL TNFα, 10 ng/mL IFNγ, 10 ng/mL CTGF) for 72 h.

On day 3, chondropellets were harvested for histology and molecular profiling. For histological analysis, chondropellets were cryosectioned and 6 μm frozen sections were stained with hematoxylin and eosin (Sigma), and 1% toluidine blue (Sigma) for semi-quantitative assessment of pellet structure and chondrogenic differentiation, respectively. For molecular profile analysis, RNA extraction from chondropellets was performed using the RNeasy Mini Kit (Qiagen, Frederick, MD, USA) according to the manufacturer’s instructions. Total RNA (1 μg) was used for reverse transcription with a SuperScript™ VILO™ cDNA synthesis kit (Invitrogen). A predesigned 88-gene human osteoarthritis array (GeneQuery™ Human Osteoarthritis and Cartilage repair qPCR Array Kit, ScienCell) was obtained using 1000 ng cDNA per culture and processed using a StepOne real-time thermocycler (Applied Biosystems, LLC). Mean values were normalized to the *ACTB* housekeeping gene; expression levels were calculated using the 2^−ΔCt^ method with a 34-cycle cutoff. Values were represented in a stacked bar plot as the relative fold change of the chondropellets with CD10High sEVs to the chondropellets without CD10High sEV treatment (reference sample, 2^−ΔCt^ = X sample/X reference sample).

The functional enrichment analysis was performed using g:Profiler (version e108_eg55_p17_9f356ae) with the g:SCS multiple testing correction method, applying a significance threshold of 0.05 [25]. The colors for different evidence codes and for log scale are described in Appendix A.

### 2.8. Mono-Iodoacetate Model of Acute Synovial/IFP Inflammation

All animal experiments were performed in accordance with relevant guidelines and regulations. The animal protocol was approved by the Institutional Animal Care and Use Committee (IACUC) of the University of Miami, USA (approval no. 21-030 LF) and conducted in accordance to the ARRIVE guidelines [26]. Twelve (#12) 10-week-old male Sprague Dawley rats (mean weight 250 g) were used. The animals were housed to acclimate for 1 week before the experiment initiation. One rat was housed per cage in a sanitary, ventilated room with controlled temperature, humidity, and under a 12/12 h light/dark cycle with food and water provided ad libitum.

Acute synovial/IFP inflammation was generated by intra-articular injection of 1 mg of mono-iodoacetate (MIA) in 50 µL of saline into the right knee. Under isoflurane inhalation anesthesia, rat knees were flexed 90° and MIA was injected into the medial side of the joint with a 27G needle using the patellar ligament and articular line as anatomical references. Four (4) days later, a single intra-articular injection of CD10High sEVs in 50 µL of Euro-Collins solution (MediaTech, Woodland, CA, USA) was performed, having as control (1) rat knees receiving MIA but not IFP-sEVs (Only MIA group) and (2) healthy rat knees. Animals were sacrificed at day 4 after IFP-sEV injection (d8 in total). This short exposure to MIA has been shown to induce inflammatory changes within the synovium and adjacent IFP [27].

### 2.9. Cytochemical Staining and Lubricin (PRG4) Immunolocalization In Situ

Rat knee joints were harvested by cutting the femur and tibia/fibula 1 cm above and below the joint line, muscles were dissected and removed, and joints were fixed with 10% neutral buffered formalin (Sigma-Aldrich) for 14 days at room temperature. Knee joints were decalcified, cut along the sagittal plane in half, embedded in paraffin, and serial 4 μm sections obtained. Toluidine blue staining was performed to evaluate the sulfated glycosaminoglycan content of articular cartilage. Masson’s trichrome staining was performed to evaluate the collagen type and content of the articular cartilage. Microscope images of cytochemically stained tissues were acquired using a x10 and x20 objective Leica DMi8 microscope with Leica X software (Leica). Histochemical staining quantitative analysis was evaluated in 3 rat knees per condition and 4 microscopy fields per knee with Fiji/ImageJ 2.14.0 software.

For anti-lubricin (anti-PRG4) immunofluorescence staining, sections were permeabilized with 1x PBS + 0.2% Triton X-100 for 20 min at room temperature and incubated with blocking buffer (1x PBS + 0.1% Triton X-100 with 1% bovine serum albumin) for 1 h at room temperature. In between different treatments, sections were washed with 1X PBS. Mouse anti-rat PRG4 monoclonal antibody (Sigma, Burlington, MA, USA) was prepared in blocking buffer (1:500) and sections were incubated at 4 °C overnight. Sections were washed with 1X PBS + 0.01% Triton X-100 and incubated for 2 h with secondary antibody Alexa Fluor647 conjugated goat anti-mouse IgG antibody (Thermo Fisher Scientific) at room temperature. Controls were incubated with secondary antibody only. All sections were rinsed with 1x PBS, mounted in prolong gold antifade reagent with DAPI (Invitrogen), and microscope images were acquired using a x20 objective Leica DMi8 microscope with Leica X software (Leica).

### 2.10. Statistical Analysis

Normal distribution of values was assessed by the Kolmogorov—Smirnov normality test. In the presence of a non-normal distribution of the data, one-way or two-way ANOVA tests were used for multiple comparisons. All tests were performed with GraphPad Prism v7.03 (GraphPad Software, San Diego, CA, USA). Level of significance was set at *p* < 0.05.

## 3. Results and Discussion

### 3.1. Distinct Molecular Profiles of IFP-MSC Based on CD10 Protein Expression Level

CD10 is a surface neutral endopeptidase expressed in multiple cell types including MSC [28,29], with enzymatic activity neutralizing various signaling substrates including substance P (SP) [30,31]. Specifically, SP is a neuropeptide associated with nociceptive pathways that is secreted by sensory nerve fibres in the synovium and IFP tissues. Upon its secretion, it actively affects local inflammatory/immune and fibrotic responses by modulation of cell proliferation, activation, and migration to sites of inflammation, and the expression of recruiting chemokines and adhesion molecules [3,4]. Importantly, for the onset and progression of human inflammatory joint diseases, SP-secreting sensory nerve fibers predominate over sympathetic fibers in anterior knee pain [7,8,32], whereas SP secretion is increased in synovial fluid during joint inflammation [3]. On this basis, we have previously reported that IFP-MSC can efficiently degrade substance P and suppress T cell proliferation in vitro, whereas upon intra-articular infusion in vivo, IFP-MSC can dramatically reverse signs of synovitis and IFP fibrosis [33,34]. Specifically, our in vivo animal studies revealed a correlation between CD10 expression magnitude in IFP-MSC and the reduction or even absence of SP+ fibers in areas of active inflammation and fibrosis for both the synovium and IFP tissues. Based on the correlation between CD10 positivity and therapeutic outcomes, levels around 70% or more should represents an effective therapeutic outcome [33].

In the present study, IFP-MSC cultured in regulation-compliant (Ch-R and hPL) media showed similar fibroblast-like morphology but importantly had different CD10 expression levels (Figure 1A). Ch-R cultured IFP-MSC had 96.5 ± 5.6% expression whereas hPL cultured IFP-MSC had 59.2 ± 15.7% expression. According to the abovementioned 70% threshold for an effective therapeutic outcome in vivo, we named Ch-R cultures as CD10High (more therapeutic) and hPL cultures as CD10Low (less therapeutic).

Molecular profiling of CD10High versus CD10Low IFP-MSC revealed that 25 out of 90 genes tested were more highly expressed in CD10High IFP-MSC, with 8 genes being more than twofold higher (*KDR*, *COL10A1*, *NGFR*, *PROM1*, *ALPL*, *EGF*, *BMP7*, *ITGAX*) (Figure 1B). Interestingly, genes tested were grouped in phenotype-related cohorts with MSC-associated, chondrogenic/osteogenic, and MSC cohorts showing overall the most prominent fold expression change between CD10High and CD10Low IFP-MSC cultures (Figure 1C). Functionally, *KDR* (*VEGFR2*) and *COL10A1* genes, which are more than 10-fold higher expressed in CD10High IFP-MSC, are strongly related to robust pro-chondrogenic and pro-angiogenic MSC actions [35,36]. Interestingly, studies show that the JNK-EGR1 signaling axis promotes TNF-α-induced endothelial differentiation of MSC via VEGFR2 expression [35]. COL10A1 is a specific marker of hypertrophic chondrocytes and is critical for endochondral bone formation, as mutation and altered *COL10A1* expression is often accompanied by abnormal chondrocyte hypertrophy in many skeletal diseases [36]. Among the highly expressed genes (more than twofold), *NGFR* (*CD271*) and *PROM1* (*CD133*) have important roles in pro-angiogenic, immunomodulatory, and chondrogenic MSC actions. In vivo studies showed that CD271^+^ MSC and CD133^+^ MSC infusion in ischemia animal models result in increased pro-angiogenic and decreased pro-inflammatory signaling [37,38]. Several studies have associated high NGFR gene and protein expressions in MSC with their increased chondrogenic capacity in vitro and osteochondral cartilage repair in vivo [39,40,41]. Pro-chondrogenic and pro-angiogenic actions can be also attributed to MSC with high expression levels of *ALPL*, *EGF*, *BMP7*, and *ITGAX* genes [42,43,44]. Of note, the *MCAM* (*CD146*) gene higher expressed in CD10High IFP-MSC is directly related to increased immunomodulatory MSC actions in vitro and in vivo [45].

In contrast, the CD10Low IFP-MSC molecular profile showed more than twofold higher expression of 45 genes compared to the CD10High IFP-MSC profile. Specifically, *FGF9*, *FGF10*, *ACAN*, and *IGF-1* genes seem to be a characteristic molecular signature for CD10Low IFP-MSC as they are expressed more than 150-fold higher, respectively. These genes are involved in proliferation and differentiation of MSC signaling. Studies have shown that *FGF9* and *FGF10* expression acts as an important regulator of early chondrogenesis, partly through the AKT/GSK-3β signaling pathway [46,47]. High *ACAN* gene expression is related to pro-chondrogenic MSC action, as its protein is a key GAG-containing proteoglycan in cartilage and plays an important role in stabilizing the ECM in articular cartilage [48]. Moreover, a recent study showed that IGF-1-transfected MSC have chondroprotective effects in an OA model [49].

Overall, CD10 protein levels in MSC strongly affected their molecular profile. CD10High IFP-MSC had a more distinct molecular profile consisting of only eight genes that were expressed significantly higher (more than twofold) compared to CD10Low IFP-MSC. Furthermore, the CD10High IFP-MSC molecular profile was related to robust pro-chondrogenic and pro-angiogenic MSC actions.

The CD10 protein levels in parental MSC were strongly correlated with the CD10 protein levels in their sEV cargo. Specifically, CD10 protein cargo was 610 ± 20.6 pg/mL for CD10High sEVs and 200 ± 20 pg/mL for CD10Low sEVs (Figure 1D).

### 3.2. CD10High and CD10Low sEVs Possess Immunomodulatory miRNA Cargo

In results similar to those of our previous study [19], upon ultracentrifugation and CD63^+^ immunoselection, both CD10High and CD10Low sEVs showed high purity for CD9 (>90%) and <200 nm sizes. From 166 MSC-related miRNAs analyzed, 154 and 151 miRNA cargos were present in CD10High (Figure 2A) and CD10Low sEVs (Figure 3A), respectively. In CD10High sEVs, nine miRNAs cargos were predominant (hsa-miR-146a, hsa-miR-4466, hsa-miR-1290, hsa-miR-6089, hsa-miR-1246, hsa-miR-3665, hsa-miR-7975, hsa-miR-4516, hsa-miR-4454) whereas in CD10Low sEVs, six miRNAs cargo (hsa-miR-146a, hsa-miR-6089, hsa-miR-4466, hsa-miR-3665, hsa-miR-4454, hsa-miR-7975) were predominant. Furthermore, two of the commonly highly present miRNAs (hsa-miR-146a and hsa-miR-6089) are potent regulators of the immune system, specifically involving T cell and macrophage activation and polarization. The first highly present miRNA cargo, hsa-miR-146a, belongs to the hsa-miR-146 family of genes that are expressed in response to pro-inflammatory stimuli through a negative feedback loop to modulate inflammation. Specifically, studies have shown that hsa-miR-146a negatively regulates adaptive immunity by modulating adaptor protein (AP)-1 activity and IL-2 expression in T cells, as well as immune cell activation and cytokine production [50,51]. The second highly present miRNA cargo, hsa-miR-6089, inhibits the activation of macrophages and regulates the generation of IL-6, IL-29, and TNF-α by directly controlling TLR4 signaling [52]. Reactome analysis of miRNAs predominant in CD10High sEVs and CD10Low sEVs showed their involvement in the regulation of six gene groups related to gene expression, the immune system, NGF/PDGF/Wnt pathways, metabolism of proteins, the cell cycle, and cellular responses to stress (Figure 2B and Figure 3B).

Overall, most detected miRNAs were commonly present in IFP-MSC sEVs; however, five miRNAs were specific for CD10High sEVs (hsa-miR-451a, hsa-miR-374a-5p, hsa-miR-525-5p, hsa-miR-499a-5p, hsa-miR-369-3p) and two miRNAs were specific for CD10Low sEVs (hsa-miR-132, hsa-miR-218-5p) (Figure 4A). Moreover, the histological profile of the synovium in OA preclinical models [17], and often in OA patients [53], is characterized by synovial lining hyperplasia and IFP fibrosis. The main cellular types contributing to these histological changes are synoviocytes and macrophages polarized towards the pro-inflammatory l M1 phenotype (reviewed in [54]). On this basis, macrophage phenotype manipulation has been proposed as a potential OA therapy given that initial data, including results from recent studies in our laboratory, indicate polarization of macrophages towards an alternative anti-inflammatory M2 phenotype and IFP fibrosis modulation [16,19].

In the present study, in silico analysis revealed a functional correlation of identified miRNAs in CD10High sEVs and CD10Low sEVs with known highly expressed target genes in macrophages and synoviocytes (Figure 4B). For CD10High sEVs, the prediction scoring system revealed *CCL2* as a target for hsa-miR-374a-5p (92% probability), *CD163* as a target for hsa-miR-369-3p (67% probability), *IL-10* as a target for hsa-miR-374a-5p (68% probability), *TIMP-2* as a target for hsa-miR-369-3p (53% probability), and *PRG4* as a target for hsa-miR-369-3p and hsa-miR-374a-5p (56% and 66%, respectively). Interestingly, hsa-miR-369-3p and hsa-miR-374a-5p CD10High sEVs cargo are potent regulators of the anti-inflammatory M2 macrophage phenotype (*CCL2*, *CD163*, *IL-10*, *TIMP-2*) [16,55,56] and lubricin protein production from synoviocytes (*PRG4*) [57]. For CD10Low sEVs, the prediction scoring system revealed *TAC1*, *ARG1*, *PRG4* as targets for hsa-miR-218-5p (84%, 67%, 80%, respectively). Of note, hsa-miR-218-5p CD10Low sEVs’ cargo is a potent regulator of inflammation/pain (*TAC1*), the anti-inflammatory M2 macrophage phenotype (*ARG1*), and lubricin protein production from synoviocytes (*PRG4*) [16,17,57]. Overall, from a clinical standpoint, the potent functionality of these miRNAs strongly supports the notion of developing novel cell-free therapeutics for inflammation/fibrosis reversal based on the CD10 signature of IFP-MSC sEVs.

### 3.3. CD10High and CD10Low sEVs’ Anti-Inflammatory Effects on Synoviocytes

Synovial lining hyperplasia is a main characteristic of IFP/synovium inflammation in OA. In general, there are two types of resident synoviocytes contributing to such pathological conditions; type A (macrophage-like synoviocytes) and type B (fibroblast-like synoviocytes) are mainly responsible for maintenance of synovial homeostasis [54,58]. Herein, we separately investigated the functional effects of CD10High and CD10Low sEVs on stimulated synoviocytes in vitro to identify the optimal therapeutic IFP-MSC sEV scheme for in vivo application.

In cocultures, both CD10High and CD10Low sEVs were internalized by TIC-primed SYNs (SYN TIC) and their proliferation was attenuated (Figure 5A,B). This finding suggests that IFP-MSC sEVs, irrespective of the CD10 levels in their parental cells, could attenuate synovial lining hyperplasia upon their in vivo intra-articular infusion. Similarly, we and others have reported that SYN exposure to MSC sEVs results in suppressed healthy-SYN [19] and rheumatoid arthritis-SYN [59] activation, migration, and invasion in vitro.

At the molecular level, SYN and TIC culture gene expression was strongly affected by both CD10High and CD10Low sEVs. From 88 genes analyzed, most expression levels were downregulated upon exposure to IFP-MSC sEVs (Figure 5C). Specifically, CD10High sEVs increased the expression level of 45 genes in SYN TIC, with only 13 genes being more than twofold higher (*PTGS2*, *MDM2*, *EREG*, *KRT7*, *IGF1*, *ESM1*, *IL15*, *ITGA4*, *VIM*, *POSTN*, *CSNK1D*, *CD14*, and *ALB*). Reactome analysis of highly expressed genes in CD10High sEVs/SYN TIC showed their involvement in the regulation of five molecular pathways related to the: regulation of cell population proliferation, cellular response to stimulus, prostaglandin biosynthetic process, regulation of neuroinflammatory response, and regulation of protein phosphorylation (Figure 5D). Also, CD10Low sEVs increased the expression levels of 19 genes in SYN TIC, with only nine genes being more than twofold higher (*PTGS2*, *EREG*, *MDM2*, *KRT7*, *NGF*, *POSTN*, *CD14*, *ESM1*, and *ITGA4*). Reactome analysis of highly expressed genes in CD10Low sEVs/SYN TIC showed their involvement in the regulation of five molecular pathways related to the regulation of cell population proliferation, regulation of protein phosphorylation, cellular response to stimulus, prostaglandin biosynthetic process, and regulation of cellular senescence (Figure 5D).

Among the commonly expressed genes, *PTGS2* (*COX2*) was the highest expressed (more than 500-fold) after IFP-MSC sEV exposure compared to SYN TIC cultures alone. Importantly, *PTGS2* plays a crucial role in the promotion of the prostaglandin E2 (PGE2) biosynthesis pathway in fibroblast-like cells (such as MSC), indicating its strong involvement in cell immunomodulatory capacity in vivo [60]. On this basis, via COX2-dependent PGE2 secretion, fibroblast-like cells can polarize macrophages to an alternative anti-inflammatory M2 phenotype, attenuate T cell proliferation and differentiation to Th17, and increase T cell differentiation to Tregs (reviewed in [60]). In parallel, a pioneering study showed a COX2-dependent PGE2-regulated suppression of fibroblast proliferation [61] that in concert with our data could explain the reduction of synoviocyte proliferation upon exposure to IFP-MSC sEVs. Among the highly expressed genes (more than 50-fold), *EREG* and *MDM2* have important pro-survival and trophic roles in synoviocytes. Studies showed that EREG (epiregulin), a ligand for epidermal growth factor receptor (EGFR), can affect EGFR signaling, which is critical for maintaining the superficial layer of articular cartilage, and preventing osteoarthritis initiation [62]. Studies showed that higher *MDM2* gene expression in fibroblast-like cells results in higher extracellular matrix accumulation (collagens and fibronectin) [63]. However, MDM2, which is a ubiquitin ligase and binds p53 transcriptional factor for the apoptotic pathway, has been associated with rheumatoid arthritis disease promotion [64]. Although these findings show only some of the pleiotropic IFP-MSC sEV actions on synoviocytes, they invite studies involving OA.

Excluding the commonly highly expressed genes, CD10High sEV-exposed SYN TICs showed additional expression of genes (such as *IGF-1* and *ALB*) involved in synovial fluid enrichment with anabolic factors for cartilage homeostasis. *IGF-1* gene expression was associated with insulin-like growth factor (IGF) signaling pathway activation that can promote chondrocyte proliferation, enhance matrix production, and inhibit chondrocyte apoptosis [65]. Also, high *ALB* gene expression is related to synoviocytes’ increased production of albumin, the most common protein in synovial fluid, which plays an important role in the lubrication of the cartilage, thereby protecting the joint cartilage surface from degradation [66]. Therefore, due to more potent and pleiotropic effects of CD10High sEVs on synoviocytes at the molecular level, we used CD10High sEVs as our platform for further in vitro and in vivo experimentation.

### 3.4. CD10High sEVs-Treated Chondrocytes Show a Chondroprotective Molecular Profile under Inflammatory Conditions In Vitro

For inflammatory joint diseases, the synovium/IFP anatomical and functional unit [67] can serve as a site for immune cell infiltration, neovascularization, and the origin of pro-inflammatory and articular cartilage (AC) catabolic molecules [11,54]. In this pathological microenvironment, chondrocytes become responsive to pro-inflammatory factors, especially tumor necrosis factor-α (TNFα) and interleukin-1β (IL-1β). These factors initiate the degradation of cartilage extracellular matrix mediated by matrix-metalloproteinases (MMPs), a disintegrin and metalloproteinase with thrombospondin motifs [11]. Herein, we simulated this pro-inflammatory microenvironment in vitro by co-culturing MSC-derived chondropellets with inflamed synoviocytes (Figure 6A).

In the current experiments, IFP-MSC were successfully induced towards chondrogenesis in 3D micromass pellet cultures for 15 days. On this basis, we acknowledge that despite MSC ability to differentiate towards the chondrogenic lineage, their terminal phenotype is reminiscent of one characteristic of endochondral bone formation rather than articular cartilage chondrocytes. However, we and others have demonstrated that on day 15, MSC are adequately differentiated and possess most of the major characteristics of chondrocytes without entering hypertrophy and mineralization in vitro [33,68,69]. Subsequently, our generated chondropellets and SYN were co-cultured with and without CD10High sEVs under inflammatory/fibrotic TIC conditions for 3 days. Chondropellet histological analysis revealed that although CD10High sEVs, treated and non-treated, have similar structure, CD10High sEV treatment resulted in better chondrocyte differentiation and sulfated glycosaminoglycan production in vitro (Figure 6B).

At the molecular level, CD10High sEV-treated chondropellets showed a distinct transcriptional signature with higher expression of 19 (*TGFB1*, *BMP7*, *WNT8A*, *COL1A2*, *GDF11*, *GADD45B*, *WNT9A*, *SOX9*, *BMP6*, *TNFSF11*, *RUNX2*, *DKK1*, *ACAN*, *COL1A1*, *MAPK8*, *WNT7A*, *NOTCH1*, *WNT4*, and *TLR2*) out of 88 genes tested compared to non-treated chondropellets (Figure 6C). Reactome analysis of highly expressed genes in CD10High sEV-treated chondropellets showed their involvement in the regulation of 10 chondroprotection- and cartilage homeostasis-related molecular pathways (Figure 6D). Specifically, seven genes demonstrated a more than twofold higher expression (*TGFB1*, *BMP7*, *WNT8A*, *COL1A2*, *GDF11*, *GADD45B*, and *WNT9A*) in CD10High sEV-treated chondropellets. Functionally, *TGFB1*, *BMP7*, and *WNT8A* genes, which were more than fivefold higher expressed, are strongly related to robust cartilage homeostasis effects. Both TGFB1 and BMP7 are major articular cartilage homeostasis growth factors as they arrest chondrocytes’ differentiation at an early stage of hypertrophy and inhibit expression of *COL10A1*, *VEGF*, *MMP13*, and *BGLAP* (osteocalcin) [43,70]. Interestingly, studies showed that canonical Wnt signaling, via Wnt8A gene expression, skews TGFB signaling in chondrocytes towards anabolic signaling via ALK1 and Smad 1/5/8 [71]. Among the more than twofold higher expressed genes, *COL1A2* and *GDF11* are strongly related to chondrocyte proliferation/differentiation and extracellular matrix synthesis [72,73]. Finally, *GADD45B* expression enhances *COL10A1* transcription via the MTK1/MKK3/6/p38 axis in terminally differentiating chondrocytes [74], whereas *WNT9A* signaling is required for joint integrity and regulation of Indian hedgehog (Ihh) during chondrogenesis [75]. Overall, our data show that CD10High sEVs possess potent chondroprotective and anabolic effects for chondrocytes in vitro.

### 3.5. CD10High sEVs Show Chondroprotective and Anabolic Effects for Articular Cartilage In Vivo

We have previously reported that functionally, IFP-MSC sEVs can significantly reduce synovitis/IFP fibrosis, and also affect macrophage polarization towards an anti-inflammatory therapeutic M2 phenotype in inflammatory conditions in vivo [19]. Herein, an acute synovial/IFP inflammation rat (MIA) model was used to test the chondroprotective capacity of CD10High sEVs upon intra-articular infusion to reverse cartilage degeneration in vivo (Figure 7A).

Collagen type II and aggrecan are two major components of the extracellular matrix of healthy articular cartilage. Alterations in cartilage extracellular matrix composition and structure are characteristics of OA. Specifically, during OA progression, the presence of degrading cartilage proteins such as matrix metalloproteinases and aggrecanases result in decreased collagen type II/aggrecan content and loss of cartilage integrity [76,77]. In the current study, the diseased group (MIA only) demonstrated strong cartilage degeneration signs by reduced staining for sulfated proteoglycans compared to healthy rat knees (Figure 7B,C, upper panels). According to the Udo et al. cartilage scoring system (0–5) for rat OA in the MIA model [27], only the diseased group developed grade 3 cartilage erosion (≤50% of joint surface). Importantly, CD10High sEV intra-articular infusion resulted in significantly (*p* < 0.05) reduced cartilage degeneration compared to the diseased group (average scores 0.4 ± 0.5 and 1.7 ± 1.1, respectively). Masson’s trichrome staining has also been used as a suitable histochemical method [78] to evaluate collagen compositional changes. In general, the diseased group showed increased cartilage discoloration (stained as red) compared to other groups. In contrast, CD10High sEV intra-articular infusion resulted in significantly (*p* < 0.05) increased collagen composition, similar to the healthy control group (Figure 7B,C, middle panels). Similar to our findings, previous studies using intra-articular infusions of MSC sEVs derived from various tissue sources, such as bone marrow, amniotic fluid, and umbilical cord, reported the anabolic effects of sEVs to enhance matrix synthesis of collagen type II and sulfated glycosaminoglycans (reviewed in [79]). However, in our study, for the first time, we demonstrate the chondroprotective effects of CD10High sEVs derived from IFP tissue.

The avascular nature of articular cartilage limits the capacity for perivascular MSC to migrate from the neighboring microenvironment (i.e., subchondral bone) to restore inflammatory and structural tissue imbalances. However, a pool of resident cartilage-derived stem/progenitor cells (CSPCs) is present in the upper zone cartilage zone and is associated with the expression of proteoglycan 4 (PRG4) encoding for proteoglycan 4 or lubricin [80,81]. Interestingly, studies showed that CPSC is an intermediate cell population between MSC and differentiated chondrocytes, which can be stimulated to induce tissue healing and homeostasis at different stages of cartilage degradation in vivo [82,83]. On this basis, we performed in situ PRG4 protein expression evaluations to determine phenotypic cellular changes promoted in articular cartilage in disease and after intra-articular CD10High sEV infusion (Figure 7B,C, bottom panels). Under these conditions, the healthy group showed a rigid distinct PRG4+ CSPC upper layer on articular cartilage. In the diseased group, PRG4 expression exhibited a reduction in the upper layer of articular cartilage, and simultaneously a significant (*p* < 0.05) increase in the intermediate zone randomly distributed within chondrocytes. This phenomenon can be attributed to the fact that intermediate zone chondrocytes increase the expression of PRG4 in order to replenish the significant loss from the upper cartilage surface during disease progression. Notably, CD10High sEV intra-articular infusion resulted in significant (*p* < 0.05) preservation of PRG4 expression on the upper cartilage surface and only minor expression from the intermediate zone chondrocytes. This effect can be attributed not only to the general CD10High sEVs’ immunomodulatory/chondroprotective miRNA profile (Figure 2), but also more specifically to the presence of hsa-miR-369-3p and hsa-miR-374a-5p miRNA cargo that strongly affected *PRG4* gene expression in target cells (Figure 4B). Therefore, CD10High sEV treatment suggests robust chondroprotective and anabolic effects on articular cartilage in vivo.

## 4. Conclusions

In summary, CD10High sEVs show potent molecular immunomodulatory and chondroprotective profiles that significantly affect synoviocyte and chondropellet functionality under inflammatory conditions in vitro. Notably, in an animal model of acute synovitis/IFP fibrosis, CD10High sEV intra-articular infusion resulted in robust chondroprotective and anabolic effects that retained articular cartilage structure/composition. However, the effects of CD10High sEVs in post-traumatic OA (PTOA) animal models with more advanced OA (structural/compositional alterations of articular cartilage) should be investigated in further experiments. Based on our findings, we propose a proof-of-concept viable cell-free CD10-based sEV alternative to MSC-based therapeutics in the treatment of inflammatory joint diseases, and perhaps OA phenotypes, where synovitis/IFP fibrosis and articular cartilage degradation are dominant.

## Figures and Tables

**Figure 1 cells-12-01824-f001:**
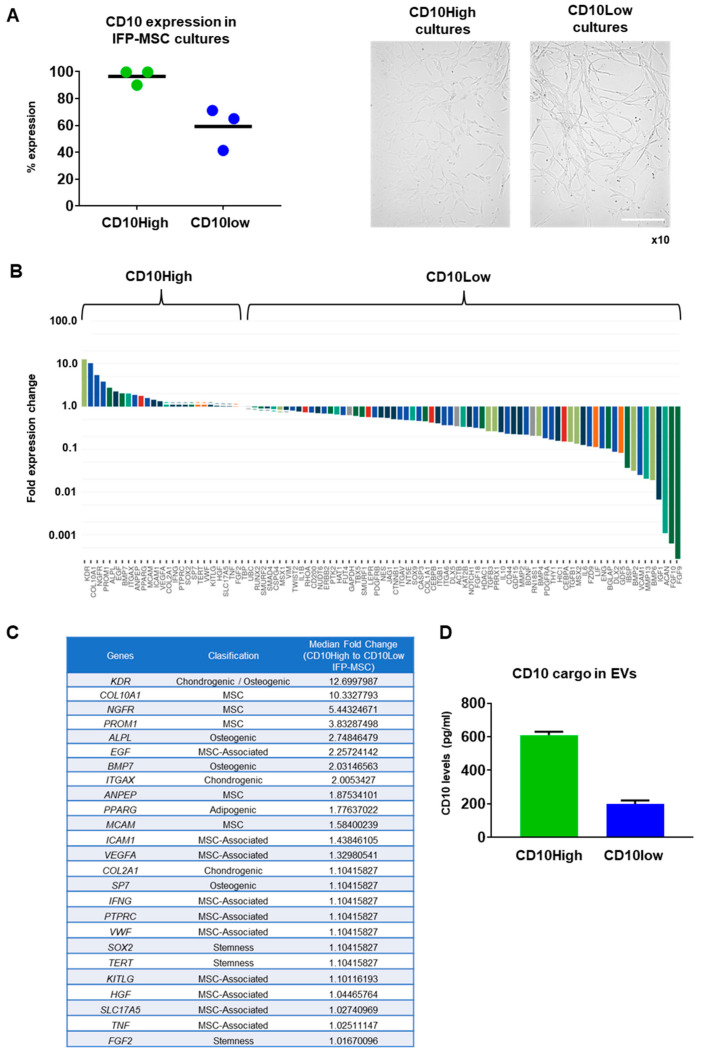
CD10High and CD10Low IFP-MSC cultures, and their molecular profiling. (**A**) IFP-MSC cultured in regulation-compliant media (Ch-R and hPL) media showed similar fibroblast-like morphology but had 96.5 ± 5.6% for Ch-R and 59.2 ± 15.7% for hPL CD10 expression levels. Magnification ×10. (**B**,**C**) Molecular profiling of CD10High versus CD10Low IFP-MSC revealed that 25 out of 90 genes tested were higher expressed in CD10High IFP-MSC, with 8 genes being more than twofold higher (*KDR*, *COL10A1*, *NGFR*, *PROM1*, *ALPL*, *EGF*, *BMP7*, *ITGAX*). Interestingly, genes tested were grouped in phenotype-related cohorts with MSC-associated, chondrogenic/osteogenic, and MSC cohorts showing overall the most prominent fold expression change between CD10High and CD10Low IFP-MSC cultures. (**D**) CD10High EVs had 610 ± 20.6 pg/mL whereas CD10Low EVs had 200 ± 20 pg/mL of CD10 protein as a cargo. Scale bar: 50 μm.

**Figure 2 cells-12-01824-f002:**
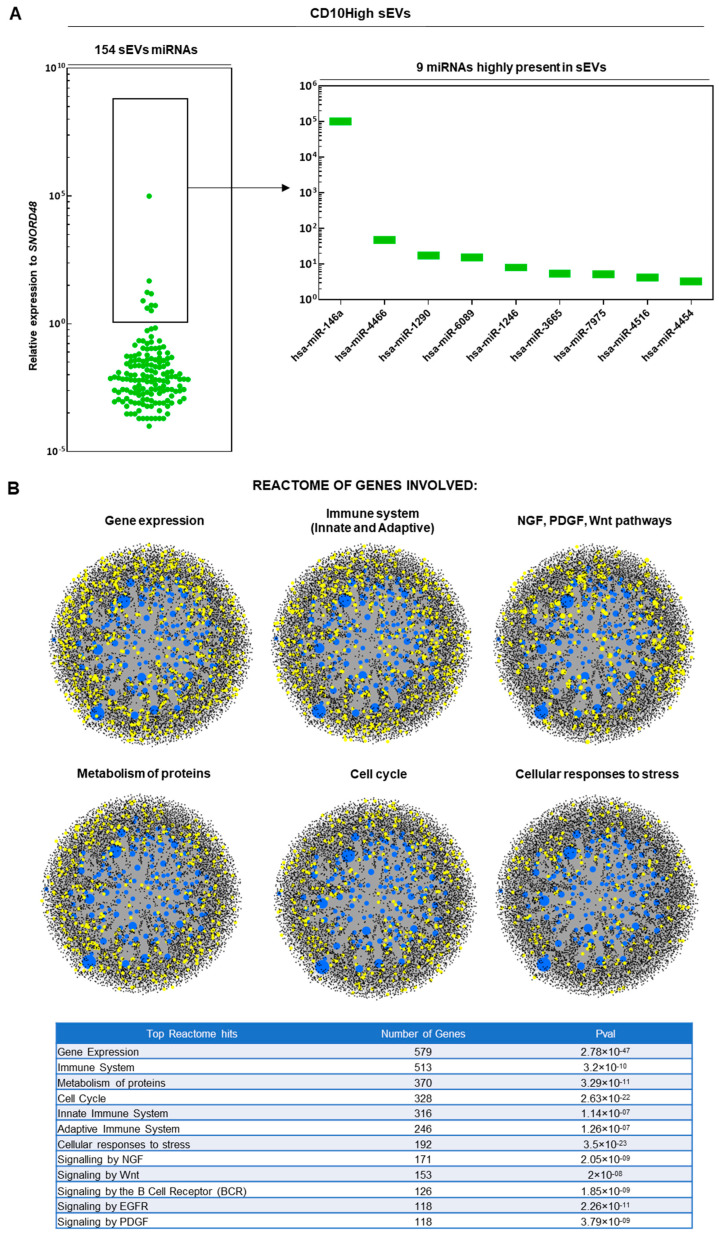
miRNA signature of CD10High sEVs. (**A**) From 166 MSC-related miRNAs analyzed, 154 miRNA cargos were present in CD10High sEVs. In CD10High sEVs, nine miRNA cargos were predominant (hsa-miR-146a, hsa-miR-4466, hsa-miR-1290, hsa-miR-6089, hsa-miR-1246, hsa-miR-3665, hsa-miR-7975, hsa-miR-4516, hsa-miR-4454). (**B**) Reactome analysis of miRNAs highly present in CD10High sEVs showed their involvement in the regulation of six gene groups related to gene expression, the immune system, NGF/PDGF/Wnt pathways, metabolism of proteins, the cell cycle, and cellular responses to stress (blue spots: miRNAs detected in sEVs, black spots: genes related to pathways, yellow spots: genes related to pathway that are affected by miRNAs detected in sEVs).

**Figure 3 cells-12-01824-f003:**
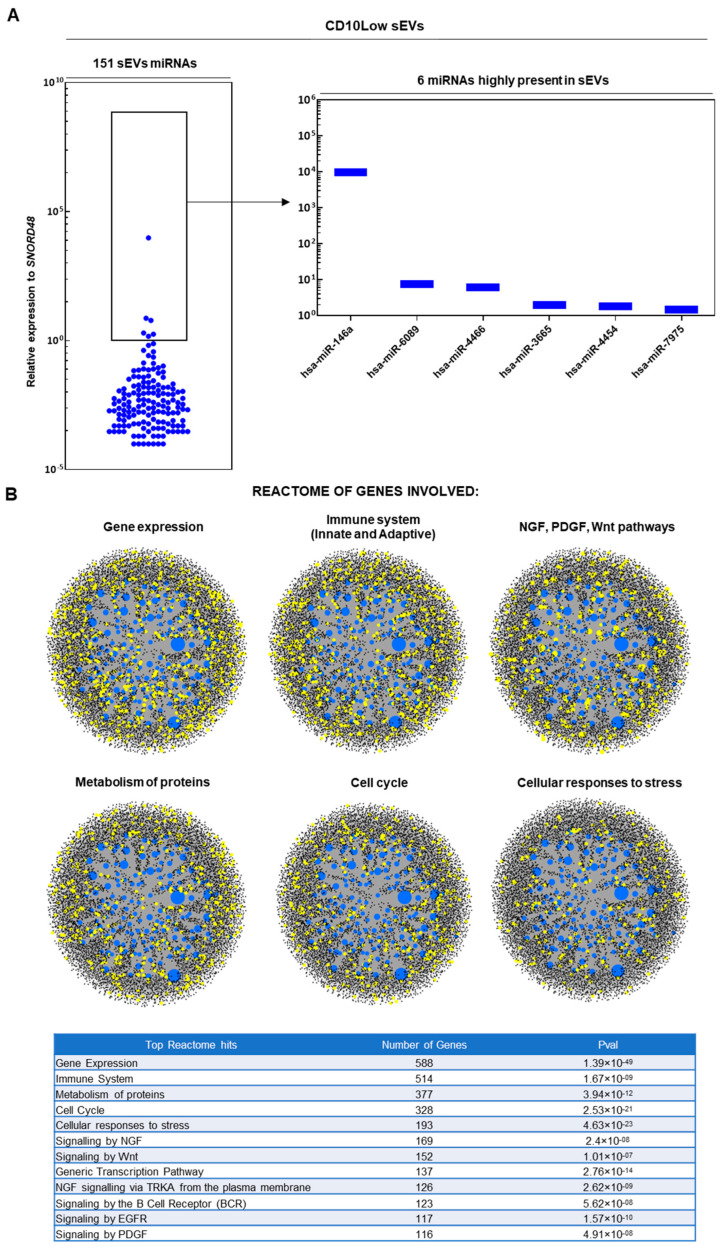
miRNA signature of CD10Low sEVs. (**A**) From 166 MSC-related miRNAs analyzed, 151 miRNA cargos were present in CD10Low sEVs. In CD10Low sEVs, six miRNAs cargo were highly present (hsa-miR-146a, hsa-miR-6089, hsa-miR-4466, hsa-miR-3665, hsa-miR-4454, hsa-miR-7975). (**B**) Reactome analysis of miRNAs highly present in CD10Low sEVs showed their involvement in the regulation of six gene groups related to gene expression, the immune system, NGF/PDGF/Wnt pathways, metabolism of proteins, the cell cycle, and cellular responses to stress (blue spots: miRNAs detected in sEVs, black spots: genes related to pathways, yellow spots: genes related to pathway that are affected by miRNAs detected in sEVs).

**Figure 4 cells-12-01824-f004:**
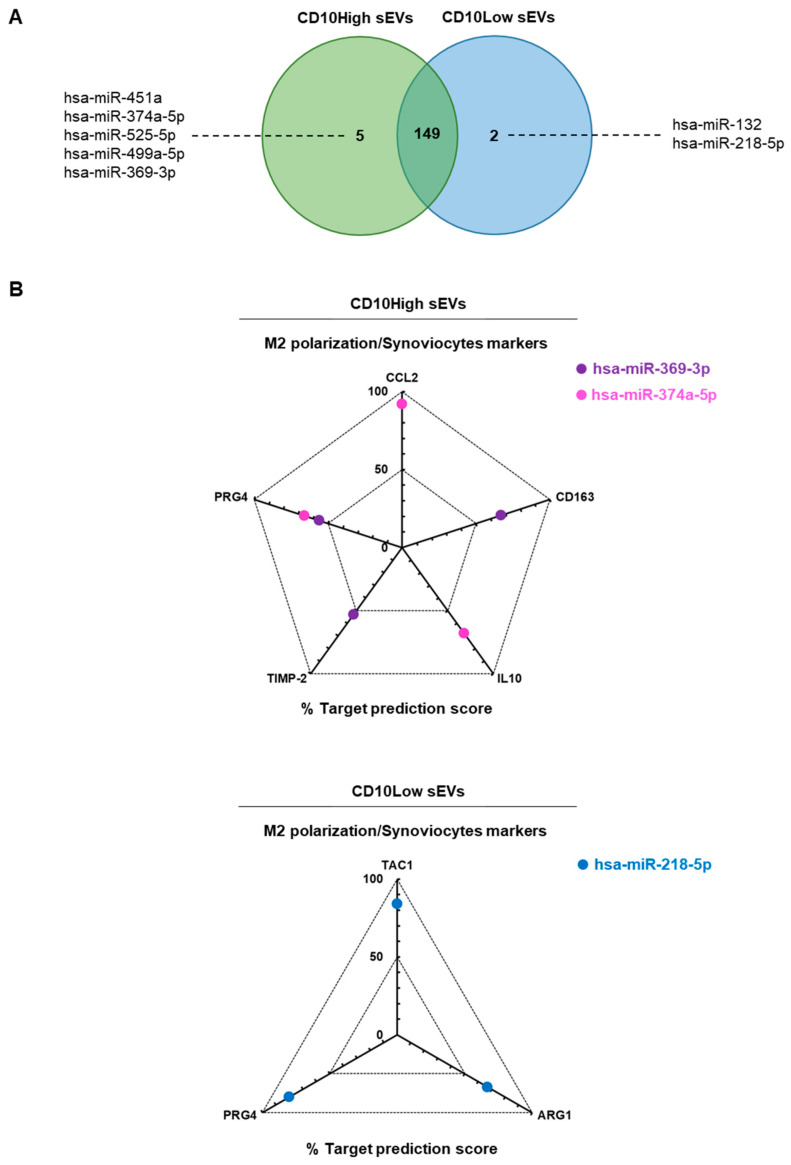
In silico analysis for functional correlation of identified miRNAs in IFP-MSC sEVs with known highly expressed target genes in macrophages and synoviocytes. (**A**) Overall, most detected miRNAs were commonly present in IFP-MSC sEVs, however, five miRNAs were specific for CD10High sEVs (hsa-miR-451a, hsa-miR-374a-5p, hsa-miR-525-5p, hsa-miR-499a-5p, hsa-miR-369-3p) and two miRNAs were specific for CD10Low sEVs (hsa-miR-132, hsa-miR-218-5p). (**B**) For CD10High sEVs, the prediction scoring system revealed *CCL2* as a target for hsa-miR-374a-5p (92% probability), *CD163* as target for hsa-miR-369-3p (67% probability), *IL-10* as a target for hsa-miR-374a-5p (68% probability), *TIMP-2* as a target for hsa-miR-369-3p (53% probability), and *PRG4* as a target for hsa-miR-369-3p and hsa-miR-374a-5p (56% and 66%, respectively). For CD10Low sEVs, the prediction scoring system revealed *TAC1*, *ARG1*, and *PRG4* as targets for hsa-miR-218-5p (84%, 67%, and 80%, respectively).

**Figure 5 cells-12-01824-f005:**
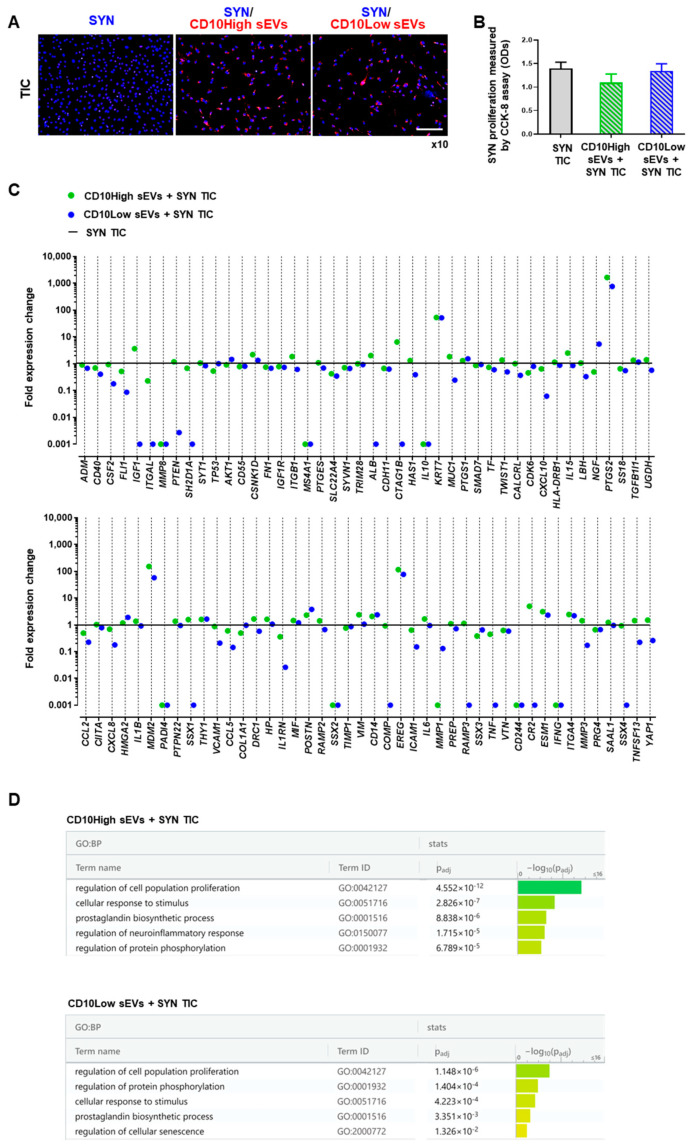
CD10High and CD10Low sEVs’ anti-inflammatory effects on synoviocytes. (**A**,**B**) Both CD10High and CD10Low sEVs were internalized by TIC-primed SYN (blue, nucleus; red, sEVs) whereas they tended to attenuate SYN TIC proliferation. (**C**) At the molecular level, SYN TIC cultures gene expression was strongly affected by both CD10High and CD10Low sEVs. Specifically, CD10High sEVs increased the expression levels of 45 genes in SYN TIC, with 13 genes more than twofold higher. Moreover, CD10Low sEVs increased the expression levels of 19 genes in SYN TIC with 9 genes more than twofold higher. (**D**) Reactome analysis of highly expressed genes in CD10High sEVs/SYN TICs and CD10Low sEVs/SYN TICs showed their involvement in the regulation of five molecular pathways. Scale bar: 50 μm.

**Figure 6 cells-12-01824-f006:**
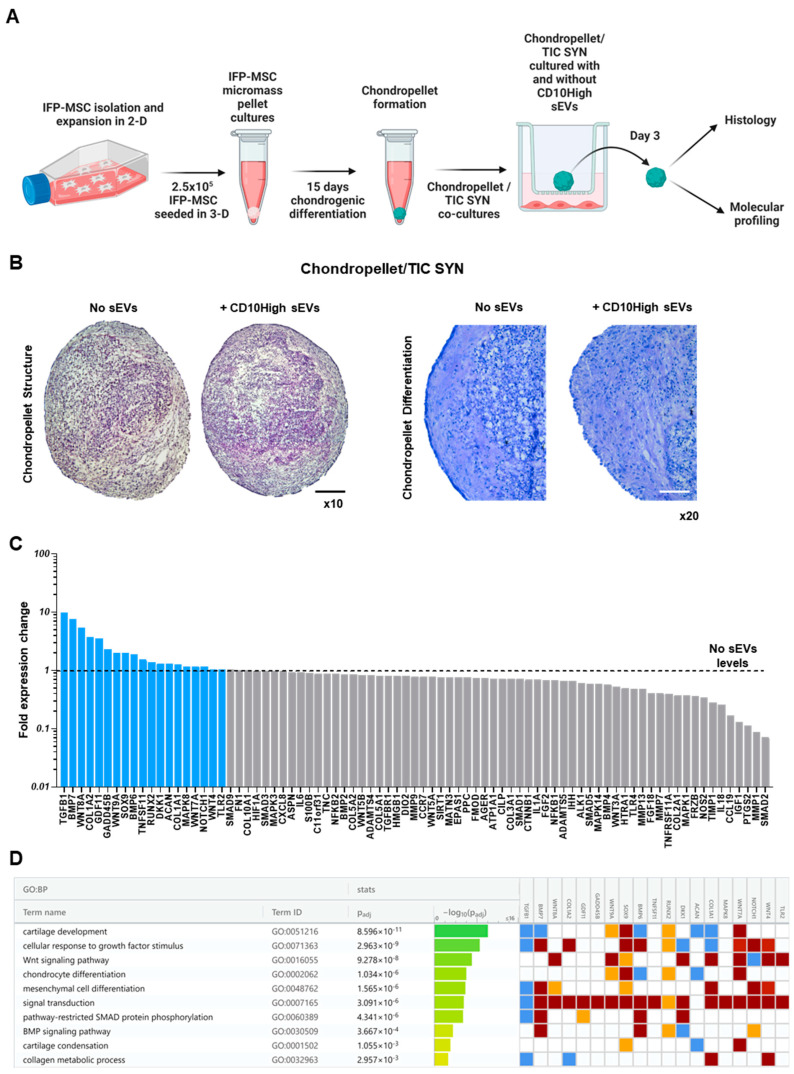
Effects of CD10High sEVs on chondrocytes in vitro. (**A**) Experimental strategy to simulate a pro-inflammatory microenvironment in vitro by co-culturing MSC-derived chondropellets with inflamed synoviocytes. (**B**) Chondropellet histological analysis revealed that although CD10High sEVs, treated and non-treated, have similar structure, CD10High sEV treatment resulted in enhanced chondrocyte differentiation and sulfated glycosaminoglycans production in vitro. (**C**) At the molecular level, CD10High sEV-treated chondropellets showed distinct transcriptional signature with higher expression of 19 genes. (**D**) Reactome analysis of highly expressed genes in CD10High sEV-treated chondropellets showed their involvement in the regulation of 10 chondroprotection- and cartilage homeostasis-related molecular pathways. Black scale bar: 50 μm, white scale bar: 25 μm.

**Figure 7 cells-12-01824-f007:**
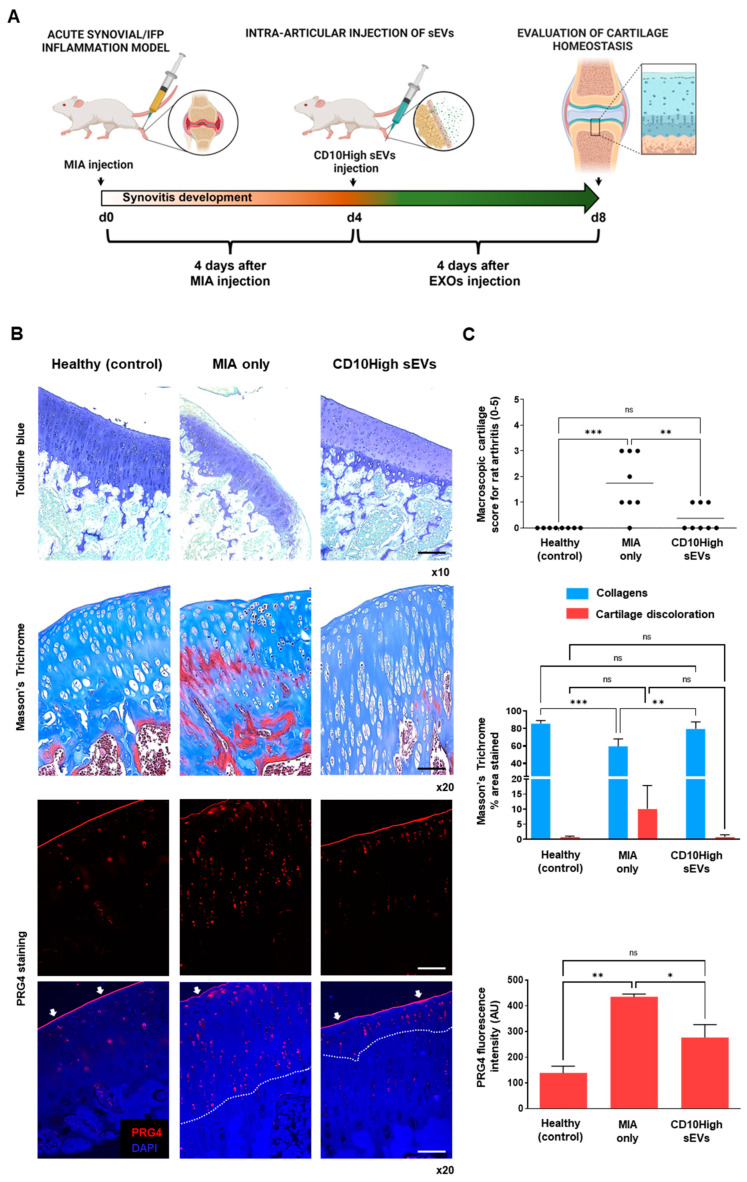
Effects of CD10High sEVs on articular cartilage homeostasis in vivo. (**A**) Schematic indicating the generation of acute synovitis/IFP fibrosis rat model, IFP-MSC sEVs’ therapeutic intervention and chronological evaluation. (**B**,**C**) Toluidine blue staining (top panel), Masson’s trichrome staining (middle panel), and PRG4 immunolocalization (lower panels) in sagittal-sectioned knees of representative rats for healthy, diseased (MIA only), or CD10High sEV treated groups. The diseased group demonstrated strong cartilage degeneration findings exemplified by reduced staining for sulfated proteoglycans. In contrast, CD10High sEV intra-articular infusion resulted in significantly reduced cartilage degeneration with only minor cartilage depressions and significantly increased collagen composition. Compared to the diseased group, the CD10High sEV group showed preservation of PRG4 expression on the upper cartilage surface and only minor expression from the intermediate zone chondrocytes (indicated by white arrows and dotted lines, ns: non-significant, * *p* < 0.05, ** *p* < 0.01, *** *p* < 0.001). Toluidine blue scale bar: 200 μm, Masson’s trichrome and PRG4 staining scale bars: 100 μm.

## Data Availability

Not applicable.

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
