# Peer review of "CD10-Bound Human Mesenchymal Stem/Stromal Cell-Derived Small Extracellular Vesicles Possess Immunomodulatory Cargo and Maintain Cartilage Homeostasis under Inflammatory Conditions"

_cells, 2023, doi:10.3390/cells12141824_

Round 1

Reviewer 1 Report

In this study, the authors focused on the CD10-expressing MSC and their exosoems. The difference in gene expression between IPF MSC with diferent expression level of CD10, microRNA expressoion profiles of their exsomes were explored, and effects of these EXOs on chondrocytes, synoviocyte and on OA animals were studied. As MSC from different or even the same tissues were heterogenous, it is of significance to characterize diferent population of MSC and clarify their function. It is of interest to clarify the therapeutic effect of CD10 expressing MSC EXOs. The overall design of the study is appropriate.

Questions and suggestion:

1. the author mentioned that the immunomudulatory function of CD10 bond MSC EXOs is important in helping maintaining cartilage homeostasis, in the study, microRNAs as well as their functional correlation with target molecules were showed, such as CCL2, CD163, IL-10, TIMP-2 for CD10High EXOs and ARG1 for CD10Low EXOs. However, these molecules were not detected in the experiment of coculture IFP MSC EXOs and SYN TIC. To clarify the immunomodulation of EXOs on synoviocytes or macrophages, it is recommended that the phenotype of these cells concerning to their polarization should be analyzed in vitro (coculture system) or in vivo.

 2. According to the results, both CD10 High EXOs and CD10 Low EXOs could help maintaing cartilage homeostasis and some different molecules may be involved, it is recommended that in the in vivo study, CD Low EXOs should be studied too.

 3. As mentioned in the introduction, CD10 play important role in the immunomodulatory effects of IFP MSC and could degrade SP. In the manuscript, the exosomes were named “CD10-bound exosomal secretome” which means CD10 was expressed on the exosomes, however, the expression level of CD10 on MSC EXOs was not showed in the results.

Author Response

REVIEWER 1:

In this study, the authors focused on the CD10-expressing MSC and their exosoems. The difference in gene expression between IPF MSC with diferent expression level of CD10, microRNA expressoion profiles of their exsomes were explored, and effects of these EXOs on chondrocytes, synoviocyte and on OA animals were studied. As MSC from different or even the same tissues were heterogenous, it is of significance to characterize diferent population of MSC and clarify their function. It is of interest to clarify the therapeutic effect of CD10 expressing MSC EXOs. The overall design of the study is appropriate.

Response: We thank the reviewer for the positive feedback and important remarks. Taking into consideration each of the reviewer’s comments, we believe we have significantly improved our manuscript by correcting typos, clarifying text, and by adding new material. Also, as requested by one of the reviewers we have now changed the nomenclature from the terms ‘exosomes’ and ‘EXOs’ with the terms ‘small extracellular vesicles’ and ‘sEVs’ throughout.

Questions and suggestion:

  1. the author mentioned that the immunomudulatory function of CD10 bond MSC EXOs is important in helping maintaining cartilage homeostasis, in the study, microRNAs as well as their functional correlation with target molecules were showed, such as CCL2, CD163, IL-10, TIMP-2 for CD10High EXOs and ARG1 for CD10Low EXOs. However, these molecules were not detected in the experiment of coculture IFP MSC EXOs and SYN TIC. To clarify the immunomodulation of EXOs on synoviocytes or macrophages, it is recommended that the phenotype of these cells concerning to their polarization should be analyzed in vitro (coculture system) or in vivo.

Response: We thank the reviewer for this valid comment. Indeed we have performed an in silico analysis of the correlation of detected miRNAs in CD10High and CD10Low EXOs with selected important genes (according to our previous experimentation and literature) that are involved in the regulation/polarization of target cells within the joint microenvironment (Figure 4). However, in the next two figures (Figures 5 and 6) we have described thoroughly the molecular profile of two major components of joint microenvironment, synoviocytes and chondrocytes, when they are exposed to CD10High and CD10Low EXOs. Specifically, for synoviocytes we discovered that CD10High EXOs increased the expression level of 45 genes in SYN TIC with 13 genes being more than two-fold higher. Moreover, CD10Low EXOs increased the expression levels of 19 genes in SYN TIC with 9 genes being more than two-fold higher. Reactome analysis of highly expressed genes in CD10High EXOs/SYN TIC and CD10Low EXOs/SYN TIC showed their involvement in the regulation of 5 molecular pathways. Of note, in this study we didn’t include another important component of the joint microenvironment, macrophages, as we wanted to focus mainly on the effects of EXOs on cartilage homeostasis. We have already performed an investigation of the effects of Crude IFP EXOs (that are not immunoselected based on CD10 but represent CD10Low EXOs) on macrophages  (doi: 10.1038/s41598-022-07569-7). We have referred to  this work in lines 69-73: ‘On this basis, further investigations revealed that IFP-MSC derived exosomes (IFP-MSC EXOs) show distinct miRNA and protein immunomodulatory profiles. Specifically, IFP-MSC EXOs infusion into the knee in an acute synovial/IFP inflammation rat model resulted in robust macrophage polarization towards an anti-inflammatory therapeutic M2 phenotype within the synovium/IFP tissues[19].’. We plan to focus on the immunomodulatory effects of CD10High and CD10Low EXOs and their involvement in macrophage polarization in a future study.   

  1. According to the results, both CD10 High EXOs and CD10 Low EXOs could help maintaing cartilage homeostasis and some different molecules may be involved, it is recommended that in the in vivo study, CD Low EXOs should be studied too.

Response: We thank the reviewer for this valid comment and we totally agree that CD10Low EXOs should also be investigated in vivo. Indeed, we have published this work in a previous study doi: 10.1038/s41598-022-07569-7. Specifically, in that study we performed a thorough investigation of the effects of Crude IFP EXOs (that are not immunoselected based on CD10 but represent CD10Low EXOs) on IFP synovitis/Fibrosis and macrophage polarization. We therefore did not include any CD10Low EXOs for the in vivo experimentation of the present study. As mentioned in the previous comment we plan to solely focus on the immunomodulatory effects of CD10High and CD10Low EXOs and their involvement in macrophages polarization in a future study.   

  1. As mentioned in the introduction, CD10 play important role in the immunomodulatory effects of IFP MSC and could degrade SP. In the manuscript, the exosomes were named “CD10-bound exosomal secretome” which means CD10 was expressed on the exosomes, however, the expression level of CD10 on MSC EXOs was not showed in the results.

Response: We thank the reviewer for this valid comment. We have now added new data with the quantitation of CD10 protein cargo levels in CD10High and CD10Low EXOs as part of Figure 1 panel D. We have added within the results section: ‘The CD10 protein levels in parental MSC are strongly correlated with the CD10 pro-tein levels in their sEVs cargo. Specifically, CD10 protein cargo was 610±20.6 pg/ml for CD10High sEVs and 200±20 pg/ml for CD10Low sEVs.’. We have also added the relevant part in Materials and Methods section to read: ‘CD10 SimpleStep ELISA kit (Abcam, MA, USA) was used to quantify the CD10 protein cargo levels (pg/ml) in IFP-MSC EXOs, following manufacturer’s instructions. Levels were determined by measuring the fluorescence (450nm) of individual samples in end-point mode (SpectraMax M5 spectrophotometer, Molecular Devices, San Jose, CA, USA). CD10 levels were normalized to total protein content per group.’    

Reviewer 2 Report

The work is really well-described and written. I think the work is really interesting in the field of cartilage homeostasis under inflammatory conditions.

Comment 1. The nomenclature of EXOS should be changed to small extracellular vesicles in all the manuscript following MISEV guidelines.

Comment 2. The authors should add scale bar in the Figure 1A, Figure 5A, Figure 6B , Figure 7B

Author Response

REVIEWER 2:

The work is really well-described and written. I think the work is really interesting in the field of cartilage homeostasis under inflammatory conditions.

Response: We thank the reviewer for the positive feedback and important remarks. Taking into consideration each of the reviewer’s comments, we believe we have significantly improved our manuscript by correcting typos, by clarifying text, and by adding new material. 

Comment 1. The nomenclature of EXOS should be changed to small extracellular vesicles in all the manuscript following MISEV guidelines.

Response: We agree with the reviewer’s comment on nomenclature. We have replaced the terms ‘exosomes’ and ‘EXOs’ with the terms ‘small extracellular vesicles’ and ‘sEVs’ throughout the text.

Comment 2. The authors should add scale bar in the Figure 1A, Figure 5A, Figure 6B , Figure 7B

Response: We have added the magnification and scale bars to the individual photomicrographs.

Reviewer 3 Report

The authors demonstrate a very relevant and interesting research analysing IFP-MSC derived EVs, highly expressing CD10 for diminishing inflammation and possessing chondroprotective properties during joint tissue diseases. I find this research paper suitable for publishing in Cells, but after some revision.

First of all, the authors present the idea of the manuscript as suitable for treating osteoarthritis (OA). However, OA is known to be less inflammation-driven disease, as compared to rheumatoid arthritis. I suggest rephrasing the concept a bit and focus on different joint diseases, not specifically OA.

Major questions:

1.     The authors presented different MSC sources, however, they did the work with IFP-MSCs, which were isolated from two male and one female donor undergoing elective knee arthroscopy. I assume the MSCs that authors used were already activated and produced those anti-inflammatory factors before isolation, so would this effect be similar with MSCs isolated from healthy donors and from different tissues? And why did the authors decide to include difference in donor sex during the study? as this might bring a lot of inconsistency between the results.

2.     The authors claim CD10 high EXO possess immunomodulatory properties only due to their miRNA profile, however, why they didn’t analyse the actual proteomic content of those EVs?

3.     Did the authors characterize their isolated EVs according to MISEV (NTA, WB, Flow, TEM?)?

104: complete medium consisting with DMEM (low glucose 1 g/L?)

114: 37C space, comma, CO2 superscript

119: Did the authors compared MSCs phenotypical surface markers, like CD90, CD73, CD105 and others?

137: mesenchymal stem cell abbreviation is missing

146: the statistical analysis is different than the one described in 281 line. The authors need to clarify the tests they used in one place.

151 says CD10-high and CD10-low MSCs. Do authors characterize and check presence of CD10 on EVs isolated from these two types of MSCs?

Also, small typos and errors are visible throughout the document, like lack of spaces between number and dimensions (1μg), superscript missing (CO2), double spaces between words. So I suggest the authors to carefully check the paper and correct small errors before publishing in a highly ranked journal – Cells.

Author Response

REVIEWER 3:

The authors demonstrate a very relevant and interesting research analysing IFP-MSC derived EVs, highly expressing CD10 for diminishing inflammation and possessing chondroprotective properties during joint tissue diseases. I find this research paper suitable for publishing in Cells, but after some revision.

First of all, the authors present the idea of the manuscript as suitable for treating osteoarthritis (OA). However, OA is known to be less inflammation-driven disease, as compared to rheumatoid arthritis. I suggest rephrasing the concept a bit and focus on different joint diseases, not specifically OA.

Response: We thank the reviewer for the positive feedback and important remarks. Taking into consideration each of the reviewer’s comments, we believe we have significantly improved our manuscript by correcting typos, clarifying text, and adding new material. Also, as requested by one of the reviewers’ we have now changed the nomenclature from the terms ‘exosomes’ and ‘EXOs’ with the terms ‘small extracellular vesicles’ and ‘sEVs’ throughout the text.

Major questions:

  1. The authors presented different MSC sources, however, they did the work with IFP-MSCs, which were isolated from two male and one female donor undergoing elective knee arthroscopy. I assume the MSCs that authors used were already activated and produced those anti-inflammatory factors before isolation, so would this effect be similar with MSCs isolated from healthy donors and from different tissues? And why did the authors decide to include difference in donor sex during the study? as this might bring a lot of inconsistency between the results.

Response: We thank the reviewer for the valid comment. Indeed, MSCs can be ‘primed’ in vivo when isolated from various pathological sites where pro-inflammatory mediators are heavily present. Herein, we isolated MSCs from IFP tissue of non-arthritic patients (elective knee arthroscopy for anterior cruciate ligament reconstruction). Therefore, MSCs derived from these IFP tissues are not exposed in vivo to a pro-inflammatory microenvironment. On this point we totally agree that it would be ideal to isolate IFP tissue from totally healthy individuals but this is ethically challenging, and if fact impossible under our IRB system. Also, we selected IFP tissue to derive our MSC cultures as this is the tissue of interest to us in order to investigate the actual sEV secretome of IFP tissue resident MSC that putatively play a crucial role in joint homeostasis. On the other hand, isolated MSCs from IFP tissues were not directly used to extract sEVs but instead they were extensively passaged (up to passage 3) in two different complete media: human platelet lysate (hPL) and chemically-reinforced (Ch-R) media. This passaging process eliminates any putative ‘in vivo priming’ effects to the actual content of isolated sEVs. We always include both male and female biological donors in order to obtain stronger data and eliminate any putative gender-related differences.  

  1. The authors claim CD10 high EXO possess immunomodulatory properties only due to their miRNA profile, however, why they didn’t analyse the actual proteomic content of those EVs?

Response: We thank the reviewer for this comment and we totally acknowledge the significance of protein cargo on sEV immunomodulatory properties. However, this is a proof-of-concept study and our first effort to characterize partially the cargo of sEVs derived from CD10High and CD10Low IFP MSC. Upon publication of this study, we plan to move further with characterizing sEV cargos and perform a very thorough analysis of their protein content. Our first try to characterize the protein cargo of sEVs derived from Crude IFP-MSC has been already published by our group (doi: 10.1038/s41598-022-07569-7). 

  1. Did the authors characterize their isolated EVs according to MISEV (NTA, WB, Flow, TEM?)?

Response: We have characterized the isolated sEVs using NTA and flow cytometry (similar to other studies published by our group doi: 10.1038/s41598-022-07569-7 and 10.3390/cells11244002. Within the Results section it reads: ‘Similar to our previous study [19], upon ultracentrifugation and CD63+ immunoselection, both CD10High and CD10Low sEVs show high purity for CD9 (>90%) and <200 nm sizes (data not shown).’

104: complete medium consisting with DMEM (low glucose 1 g/L?)

Response: It has been corrected and now reads: ‘Enzymatic digestion was inactivated with complete media with DMEM low glucose (1g/L) GlutaMAX (ThermoFisher Scientific, Waltham, MA, USA) containing 10% fetal bovine serum (FBS; VWR, Radnor, PA, USA)’

114: 37C space, comma, CO2 superscript

Response: It has been corrected and now reads: ‘All MSC were cultured at 37°C, 5% (v/v) CO2

119: Did the authors compared MSCs phenotypical surface markers, like CD90, CD73, CD105 and others?

Response: We thank the reviewer for the valid comment. Indeed, the immunophenotype of the cells is important. However we have extensively studied the immunophenotype of our isolated IFP-MSC batches and published these findings (doi: 10.1038/s41598-019-47391-2, 10.1177/0363546520917699, 10.1016/j.jcyt.2020.06.007, 10.1186/s13287-020-02107-6). Herein, in order not to repeat the same experiments we focused on the characterization of an extensive molecular profile for CD10High and CD10Low IFP-MSC. This molecular profiling yielded interesting data by showing that these two IFP-MSC subpopulations are molecularly different. On this basis, interestingly genes tested were grouped in phenotype-related cohorts with MSC-associated, Chondrogenic/Osteogenic, and MSC cohorts showing overall the most prominent fold expression change between CD10High and CD10Low IFP-MSC cultures (Figure 1).

137: mesenchymal stem cell abbreviation is missing

Response: It has been corrected and now reads: ‘A pre-designed 90 gene Taqman-based MSC qPCR array (Stem Cell Technologies, Supplementary Table S1)’              

146: the statistical analysis is different than the one described in 281 line. The authors need to clarify the tests they used in one place.

Response: We thank the reviewer for the comment. We have now totally removed statistical analysis from line 146 and kept only the explanation of statistical analysis in the separate paragraph of materials and methods section.

151 says CD10-high and CD10-low MSCs. Do authors characterize and check presence of CD10 on EVs isolated from these two types of MSCs?

Response: We thank the reviewer for this valid comment. We have now added new data with the quantitation of CD10 protein cargo levels in CD10High and CD10Low EXOs as part of the Figure 1 panel D. We have added within the results section the text: ‘The CD10 protein levels in parental MSC are strongly correlated with the CD10 pro-tein levels in their sEVs cargo. Specifically, CD10 protein cargo was 610±20.6 pg/ml for CD10High sEVs and 200±20 pg/ml for CD10Low sEVs.’. We have also added the relevant part in Materials and Methods section and it reads: ‘CD10 SimpleStep ELISA kit (Abcam, MA, USA) was used to quantify the CD10 protein cargo levels (pg/ml) in IFP-MSC EXOs, following manufacturer’s instructions. Levels were determined by measuring the fluorescence (450nm) of individual samples in end-point mode (SpectraMax M5 spectrophotometer, Molecular Devices, San Jose, CA, USA). CD10 levels were normalized to total protein content per group.’    

Also, small typos and errors are visible throughout the document, like lack of spaces between number and dimensions (1μg), superscript missing (CO2), double spaces between words. So I suggest the authors to carefully check the paper and correct small errors before publishing in a highly ranked journal – Cells.

Response: We have carefully checked the text for any typos.

Round 2

Reviewer 1 Report

The revised manuscript is well organized and the questions have been properly answered.